# Combined Coronary CT-Angiography and TAVI-Planning: A Contrast-Neutral Routine Approach for Ruling-Out Significant Coronary Artery Disease

**DOI:** 10.3390/jcm9061623

**Published:** 2020-05-27

**Authors:** Robin F. Gohmann, Philipp Lauten, Patrick Seitz, Christian Krieghoff, Christian Lücke, Sebastian Gottschling, Meinhard Mende, Stefan Weiß, Johannes Wilde, Philipp Kiefer, Thilo Noack, Steffen Desch, David Holzhey, Michael A. Borger, Holger Thiele, Mohamed Abdel-Wahab, Matthias Gutberlet

**Affiliations:** 1Department of Diagnostic and Interventional Radiology, Heart Center Leipzig, Strümpellstr. 39, 04289 Leipzig, Germany; Patrick.Seitz@helios-gesundheit.de (P.S.); Christian.Krieghoff@helios-gesundheit.de (C.K.); Christian.Luecke@helios-gesundheit.de (C.L.); Sebastian.Gottschling@helios-gesundheit.de (S.G.); Matthias.Gutberlet@helios-gesundheit.de (M.G.); 2Medical Faculty, University of Leipzig, Liebigstr. 27, 04103 Leipzig, Germany; 3Department of Cardiology, Heart Center Leipzig, University of Leipzig, Strümpellstr. 39, 04289 Leipzig, Germany; Philipp.Lauten@medizin.uni-leipzig.de (P.L.); Johannes.Wilde@medizin.uni-leipzig.de (J.W.); Steffen.Desch@medizin.uni-leipzig.de (S.D.); Holger.Thiele@medizin.uni-leipzig.de (H.T.); Mohamed.Abdel-Wahab@medizin.uni-leipzig.de (M.A.-W.); 4Institute of Medical Informatics, Statistics and Epidemiology (IMISE), University of Leipzig, Härtelstr. 16-18, 04107 Leipzig, Germany; meinhard.mende@zks.uni-leipzig.de; 5Leipzig Heart Institute, Russenstr. 69a, 04289 Leipzig, Germany; Stefan.Weiss@leipzig-heart.de (S.W.); Michael.Borger@medizin.uni-leipzig.de (M.A.B.); 6Department of Cardiac Surgery, Heart Center Leipzig, University of Leipzig, Strümpellstr. 39, 04289 Leipzig, Germany; Philipp.Kiefer@medizin.uni-leipzig.de (P.K.); Thilo.Noack@medizin.uni-leipzig.de (T.N.); David.Holzhey@medizin.uni-leipzig.de (D.H.)

**Keywords:** aortic stenosis, computed tomography coronary angiography, coronary angiography, coronary artery disease, transcatheter aortic valve implantation, diagnostic accuracy

## Abstract

**Background**: Significant coronary artery disease (CAD) is a common finding in patients undergoing transcatheter aortic valve implantation (TAVI). Assessment of CAD prior to TAVI is recommended by current guidelines and is mainly performed via invasive coronary angiography (ICA). In this study we analyzed the ability of coronary CT-angiography (cCTA) to rule out significant CAD (stenosis ≥ 50%) during routine pre-TAVI evaluation in patients with high pre-test probability for CAD. **Methods**: In total, 460 consecutive patients undergoing pre-TAVI CT (mean age 79.6 ± 7.4 years) were included. All patients were examined with a retrospectively ECG-gated CT-scan of the heart, followed by a high-pitch-scan of the vascular access route utilizing a single intravenous bolus of 70 mL iodinated contrast medium. Images were evaluated for image quality, calcifications, and significant CAD; CT-examinations in which CAD could not be ruled out were defined as positive (CAD^+^). Routinely, patients received ICA (388/460; 84.3%; Group A), which was omitted if renal function was impaired and CAD was ruled out on cCTA (Group B). Following TAVI, clinical events were documented during the hospital stay. **Results**: cCTA was negative for CAD in 40.2% (188/460). Sensitivity, specificity, PPV, and NPV in Group A were 97.8%, 45.2%, 49.6%, and 97.4%, respectively. Median coronary artery calcium score (CAC) was higher in CAD^+^-patients but did not have predictive value for correct classification of patients with cCTA. There were no significant differences in clinical events between Group A and B. **Conclusion**: cCTA can be incorporated into pre-TAVI CT-evaluation with no need for additional contrast medium. cCTA may exclude significant CAD in a relatively high percentage of these high-risk patients. Thereby, cCTA may have the potential to reduce the need for ICA and total amount of contrast medium applied, possibly making pre-procedural evaluation for TAVI safer and faster.

## 1. Introduction

Transcatheter aortic valve implantation (TAVI) has become the standard method to treat patients with severe aortic stenosis who cannot undergo surgical aortic valve replacement and has evolved into an established option in patients who can undergo surgery [1,2,3,4]. Patients currently treated with TAVI are commonly elderly, frail, and have a high prevalence of co-morbidities including chronic kidney and coronary artery disease (CAD) [3,4]. The latter is recommended to be assessed by invasive coronary angiography (ICA) prior to TAVI. Alternatively, the European Society of Cardiology (ESC), European Association for Cardio-Thoracic Surgery (EACTS), European Society of Cardiovascular Radiology (ESCR), and the Society of Cardiovascular Computed Tomography (SCCT) guidelines recommend considering coronary CT-angiography (cCTA) in certain cases [1,5,6,7].

Iodinated contrast medium (ICM) and its amount are risk factors for acute kidney injury (AKI), particularly when applied intra-arterially and to a lesser degree when applied intravenously, which negatively affects clinical outcomes [8,9,10].

Nonetheless, contrast-enhanced CT is essential for accurate anatomical characterization of the aortic root anatomy and vascular access route, and has helped to substantially reduce peri-interventional complications and mortality since TAVI was first introduced [1,5,7,11]. Even though pre-procedural CT requires ICM, it helps to identify potential peri-procedural risks, makes the TAVI-procedure more efficient and has, thus, become an integral part of the pre-procedural evaluation, also helping to reduce the amount of ICM needed during implantation [11,12].

In addition, evidence has recently emerged that cCTA could also be used for the assessment of CAD in the high-risk group of TAVI-candidates [13,14,15,16,17,18,19,20,21,22,23]. Several studies have shown cCTA to be both safe and effective in patients screened for TAVI – especially for ruling out significant CAD given its high negative predictive value (NPV) [1,5,6,7,24]. Regardless, and despite TAVI being an established procedure with 24.808 annual entries into the STS/ACC TVT Registry in 2015, an estimated 107.000 TAVI prostheses sold in 2017 and 971 procedures performed in 2019 in our institution alone, the experience in evaluation of CAD with cCTA is still very limited in this cohort with less than 2.000 patients reported on so far [25,26]. Adding CAD assessment to the pre-procedural CT protocol appears clinically attractive, but generally may require specific patient preparation and selection, adjusted scanning protocols, and increased amount of ICM.

In the present study we analyzed the accuracy and safety of cCTA acquired during pre-procedural CT as a primary screening tool for significant CAD in a large cohort of unselected TAVI patients without any patient-specific adjustments of scan parameters or patient preparation. For this we compared the diagnostic performance of cCTA and ICA and analyzed clinical events.

## 2. Methods

### 2.1. Study Design and Patient Population

In total, 517 consecutive patients were referred for pre-TAVI CT-evaluation and examined using the same CT-scanner between June 2018 and January 2019 (Figure 1). Patients routinely received ICA, which was omitted at the discretion of the local heart team if renal function was impaired and CAD could be effectively ruled out on cCTA.

Forty-four patients were excluded because of prior coronary artery bypass grafting (CABG) (*n* = 37) or non-retrospectively gated CT-acquisition (*n* = 7) (Figure 1). Another 13 patients were excluded as CAD could not be ruled out on cCTA and ICA was not performed within 3 months of the CT-examination (*n* = 5) or quantitative assessment of coronary arteries (QCA) (*n* = 8) was not possible with the available data (Figure 1).

Ultimately, 460 patients were included with 388 patients (84.3%) receiving ICA within 3 months of the CT-examination (Group A) and 72 patients (15.7%) not undergoing ICA (Group B) (Figure 1).

This study was conducted in compliance with the Declaration of Helsinki (Medical Association 2013). The local ethics committee approved the study and written informed consent was waived (reference number: 435/18-ek).

### 2.2. CT Acquisition and Image Reconstruction

All examinations were performed with the same second-generation dual-source CT-scanner (Somatom Definition Flash; Siemens, Erlangen, Germany). Scout-views of the thorax, abdomen, and pelvis were acquired anteroposteriorly and laterally for planning and positioning. First, an ECG-gated non-enhanced scan of the heart at 120 kV was acquired for coronary artery calcium scoring (CAC). This was followed by the injection of a triphasic contrast bolus through a peripheral venous catheter (20 gauge or larger) at the upper limb. The bolus consisted of 50 mL ICM (370 mg iodine/mL iopromide; Ultravist, Bayer, Leverkusen, Germany) at an injection rate of 5 mL/s, 40 mL diluted contrast (1:1; contrast - physiological saline) at 3 mL/s and 40 mL saline chaser at 5 mL/s. A retrospectively ECG-gated helical scan of the heart from caudal to cranial (scan time: 4–6 s) was automatically initiated when a threshold of 100 HU was reached in the ascending aorta. This was immediately followed by a non-ECG-gated high-pitch-scan (pitch: 3.2) in the opposite direction of the thorax, abdomen, and pelvis for access route depiction (scan time: approximately 2s). Further scan settings are listed in Appendix A.

Images were reconstructed and evaluated in standard technique as described in Appendix A using a dedicated post-processing workstation (syngo.via VB40A, Siemens, Erlangen, Germany). As a surrogate for coronary arterial density including stents, CAC was quantified using the same workstation analogous to the Agatston method [27].

### 2.3. Analysis of cCTA Data

The datasets of Group A were analyzed retrospectively by two radiologists with 3 and 6 years of experience in reading cCTA (P.S.; R.G.). Readers were blinded to the results of ICA. Coronary segments were defined according to the 18-segment coronary model [28]. All segments with a diameter of >1.5 mm were analyzed and a separate semi-quantitative score was given for calcium and artifacts:0 = no calcifications/artifacts1 = mild calcifications/artifacts2 = moderate calcifications/artifacts3 = extensive calcifications covering ≥ 50% of lumen/artifacts rendering the lumen not evaluable

A significant luminal narrowing of ≥ 50% in diameter was scored visually as present or absent. If calcifications or artifacts rendered the lumen non-evaluable, the segment was defined as positive in regard to CAD (CAD^+^). Additionally, per vessel and per patient gradings were generated by considering the respective maximum.

Separate subjective semi-quantitative scores for contrast opacification and overall image quality were given per patient:0 = non-diagnostic contrast opacification/image quality1 = diagnostic contrast opacification/image quality2 = good contrast opacification/image quality3 = excellent contrast opacification/image quality

Quantitatively, CAC, contrast opacification (attenuation at aortic bulb), and contrast to noise ratio ($\mathrm{CNR})=\frac{\mathrm{attenuation}\;\mathrm{at}\;\mathrm{aortic}\;\mathrm{bulb}\;-\;\mathrm{attenuation}\;\mathrm{at}\;\mathrm{intraventricular}\;\mathrm{septum}}{\mathrm{noise}\;\mathrm{of}\;\mathrm{subcutaneous}\;\mathrm{adipose}\;\mathrm{tissue}}$ were measured for all patients.

### 2.4. Invasive Coronary Angiography

ICA was conducted in standard technique and read by a cardiologist with 9 years of experience in coronary interventions (P.L.) blinded to cCTA [29]. If a stenosis was identified, it was graded with QCA (syngo.via VB40A, Siemens, Erlangen, Germany) using two orthogonal views [13,14,15,17,19,20]. A diameter-based stenosis of ≥ 50% was considered to be significant [13,14,15,16,17,18,19,20,23,30].

### 2.5. Clinical Events

Major adverse cardiovascular and cerebrovascular events (MACCE) were defined as the sum of all-cause mortality, cerebrovascular events, and myocardial infarction with a maximum of one event per patient [31]. MACCE in addition to acute kidney injury (AKI) were documented for the duration of the entire hospital stay only for patients having undergone TAVI.

### 2.6. Diagnostic Performance of cCTA in the Literature

The literature of the past 10 years was screened for similar studies. Inclusion criteria were CT-protocols allowing for robust depiction of coronary arteries, a cut-off of 50% for CAD, results reported per patient and ICA serving as the standard of reference. Nine studies fitting those criteria were identified. In addition to our results and those in the literature, inferential statistics are given for the aggregate of all results and are presented in a table as part of the discussion.

### 2.7. Statistical Analysis

Categorical variables are reported as counts and percentage; continuous variables are reported as mean and standard deviation or median and interquartile range for skewed distribution. For group comparisons, either Pearson’s χ^2^ or Fisher’s exact test was used for categorical variables; for ordinal values, Wilcoxon rank sum test; and for continuous variables a t-test, Welch or Wilcoxon signed-rank test (skewed distribution) was used.

For evaluation of the possible influence of scan demographics on accuracy of cCTA, multiple logistic regression analysis for correct classification according to cCTA (true positive and true negative) was applied. Initially, 11 variables (heart rate variability; heart rate; CNR; CAC; number of calcified lesions; CAC/number of calcified lesions; body mass index (BMI); subjective image quality; artifacts; subjective contrast; subjective calcium) were entered and excluded in a stepwise fashion using the Akaike information criterion. The remaining variables BMI and heart rate were fitted into Model 1a; a separate model (Model 1b) was fitted to demonstrate the sole influence of BMI. For testing the influence of calcium, calcium derived parameters (CAC, number of calcified lesions, and CAC per lesions) were added to BMI and heart rate in Model 2. Odds ratios were estimated with 95% confidence intervals. Lastly, CAC was plotted against classification according to cCTA against the reference standard ICA with QCA.

Data curation and computation of inferential statistics were done via spreadsheets (Microsoft Excel, version 2010, Microsoft Corporation, Redmond, USA). For further statistical analyses, R (version 3.4.3, R Foundation for Statistical Computing, Vienna, Austria) was used. Significance level for two tailed testing was defined as 5%.

## 3. Results

### 3.1. Baseline Characteristics

The included 460 patients had a mean age of 79.6 ± 7.4 years and 51.7% were women. Previous myocardial infarction (A: 12.6%; B: 1.4%; *p* = 0.005) and percutaneous coronary intervention (A: 28.6%; B: 5.6%; *p* < 0.001) were significantly more frequent in Group A. Concomitantly, New York Heart Association (NYHA)-classification III/IV was more frequent and left ventricular ejection fraction lower in Group A. Female gender (A: 49.2%; B: 65.3%; *p* = 0.01) was significantly more frequent in Group B. All other baseline characteristics were not significantly different between the groups (Table 1).

### 3.2. Scan Demographics

Scan demographics are listed in Table 2. Notably, median CAC was significantly higher in Group A (A: 859 [285–1975]; B: 99 [0–303]; *p* < 0.001), and within Group A significantly higher in CAD^+^-patients (CAD^+^: 1355.5 [0–10142]; CAD^−^: 226.4 [0–3641]; *p* < 0.001). All other parameters did not differ significantly between groups.

### 3.3. Prevalence of CAD

The prevalence of significant CAD (diameter stenosis of ≥ 50% on ICA) was 35.6% (138/388). The left anterior descending (LAD) was most commonly affected (47.1%, 65/138) and the most commonly affected segment was no. 7 (mid-LAD).

### 3.4. Coronary Arteries on cCTA

Subjectively rated overall-image quality of cCTA (CAD^−^: 2.17 = good-excellent; CAD^+^: 1.69 = diagnostic-good; *p* < 0.001) and contrast opacification (CAD^−^: 2.36; CAD^+^: 2.18; *p* < 0.05) were rated higher in CAD^−^-patients; calcifications and artifacts were rated as less pronounced in CAD^−^-patients (*p* < 0.05) (Table 3).

In Group A, 5636 coronary segments were identified on cCTA, of which 689 segments had a diameter of < 1.5 mm, leaving 4947 coronary segments for inclusion (Table 4). Altogether, 90.1% of coronary segments were depicted diagnostically; 490/4947 segments (9.9%) were rated non-diagnostic because of insufficient contrast opacification or artifacts (75/490), e.g. motion, beam hardening or photon starvation not originating from the coronary artery and/or heavy calcifications (415/490), limiting the exclusion of significant stenosis.

Overall, 96/388 studies (24.7%) in Group A were rated not fully evaluable or non-diagnostic because of at least one segment with severe artifacts, insufficient contrast opacification, stents or heavy calcifications impeding the exclusion of CAD and thus defined as CAD^+^. Of these 38/96 (39.6%) were possibly preventable (insufficient contrast opacification, motion artifacts).

### 3.5. Diagnostic Performance of cCTA

In Group A 80.6% of segments (3989/4947) were correctly rated CAD^−^ (Table 4); false positive ratings were given in 702 segments (14.2%). Significant stenoses were correctly identified in 222/4947 segments (4.5%); underestimation of stenosis on cCTA was observed in 34/4947 segments (0.7%) (Table 4).

Sixty-three percent of vessels (980/1551) were correctly rated as negative; false positive ratings were given in 366 vessels (23.6%). Significant stenoses were correctly identified in 189/1551 vessels (12.2%); underestimation of stenoses was observed in 16 vessels (1.0%) (Table 4).

Per patient, a negative rating for CAD was given correctly in 113/388 (29.1%); stenoses were underestimated on cCTA in 3 patients being located in segment 7 (mid-LAD), segment 2 (mid-right coronary artery) and segment 13 (obtuse marginal 1). Hence, sensitivity, specificity, positive predictive value, NPV and accuracy were 97.8%, 45.2%, 49.6%, 97.4%, and 63.9%, respectively (Table 4).

Overall, 40.2% of all patients (185/460) were read as negative for significant CAD, having excluded three false negative patients.

### 3.6. Baseline Characteristics, Scan Demographics and Accuracy of cCTA

In the multivariate analysis only heart rate and BMI were associated with correct classification of patients according to cCTA (Model 1a, Table 5). BMI had the strongest effect on accuracy of cCTA with a decreased probability of correct classification of roughly 0.82 for every 5 kg/m^2^ (Model 1b; Table 5).

Model 2 shows the association of 3 calcium-related parameters with correct classification. We observed clinically relevant association of accuracy and cCTA with absolute _log_CAC and CAC per lesion (OR = 1.64 and OR = 0.49). However, large variability causes nonsignificance (Table 5). Additionally, Figure 2 displays that extent of CAC and categorization according to cCTA (false negative, false positive, true negative, true positive) overlap extensively, indicating that CAC alone lacks discriminatory power to predict, whether or not cCTA will be able to rule out significant CAD.

### 3.7. Group Differences in MACCE and AKI

All included patients received a CT for pre-TAVI evaluation. However, after the heart team’s decision not all patients underwent a TAVI-procedure (A: 70.4%; B: 80.6%). Only for this subgroup were MACCE and AKI documented (Table 6).

Overall, frequency of clinical events was similar in both groups and ultimately no statistically significant differences were observed. With the exception of MACCE (A: 6.2%; B: 6.9%; *p* = 0.56), mainly caused by cerebrovascular events, clinical events were numerically less frequent in Group B during the hospital stay (Table 6).

When the analysis was confined to CAD^−^ patients, clinical events were also not significantly different between Group A (*n* = 73) and B (*n* = 58). Overall MACCE were numerically more frequent in Group B (A: 1.4%; B: 6.9%; *p* = 0.14), caused by cerebrovascular events (A: 1.4%; B: 5.2%; *p* = 0.27) and all-cause mortality (A: 0.0%; B: 1.7%; *p* = 0.43); AKI was numerically less frequent in Group B (A: 8.2%; B: 6.9%; *p* = 0.77).

## 4. Discussion

The impact of CAD on outcome after TAVI is unclear and remains a controversial topic in itself – being still actively debated. Nevertheless, diagnosis of CAD prior to TAVI remains part of the routine pre-TAVI work-up for several reasons, including the overlap of risk factors and symptomatology between both disease entities, and the necessity to exclude and potentially treat severe proximal disease prior to TAVI according to current guideline recommendations [1,6]. Regardless of TAVI being an established procedure for the treatment of severe aortic stenosis, the experience of excluding CAD with cCTA in this patient group is still limited, with less than 2.000 patients reported on in the literature so far.

The diagnostic performance of cCTA in our study is comparable to or even higher than reported in the literature (Table 7) [13,14,15,16,17,18,19,20,23], even though a considerable proportion of our cohort did not undergo ICA after negative cCTA, leaving only the more severely diseased and more challenging patients for correlation. At the same time, our patients had comparable risk-factors, calcium burden, and ultimately a comparable prevalence of significant CAD (35.6%); with only Harris et al. and Opolski et al. reporting on a cohort with substantially higher prevalence (Table 7) [13,15,16,17,18,19,20,23]. In contrast to other studies, no exclusion of patients with known CAD treated with stents [20,23], of technically challenging patients (e.g. based on high heart rate or arrhythmia) [13,19,24], or of patients with potentially preventable errors during acquisition (e.g. poor contrast opacification; 7% of initial cohort) [18] took place. By doing so, we intended to underline the capacity of the method and the robustness of the chosen protocol for ruling out significant CAD in this high-risk group and to make our results more easily applicable to everyday practice.

One of the main challenges for cCTA in TAVI-patients is calcium burden, which enormously reduces accuracy and is responsible for the particularly high rate of false positive results [13,14,18,20,21,23,33,34,35], and may also present challenges in extremely well contrasted examinations (Figure 3). Rossi et al. reported that the diagnostic accuracy and probability to rule out CAD by cCTA decreases at CAC ≥ 400 and Anonni et al. discuss using this cut-off for deciding whether or not to attempt the assessment of coronary arteries, whereas other studies discourage using such arbitrary thresholds [14,20,36,37].

As expected, we had a significant proportion of false positive results but failed to delineate an apparent area where a meaningful threshold in regard to CAC could have been applied to our cohort (Figure 2). Furthermore, CAC and other calcium-related parameters had no significant predictive value for correct classification of patients according to cCTA. Interestingly enough, and not well described in the recent literature, we found a negative association of BMI and also to a lesser extent of heart rate on the correct classification of cCTA. However, these relationships were weak and the relationship of baseline characteristics and scan demographics, their effect on image quality, and ultimately, the ensuing predictive capacity of cCTA is likely much more complex. Therefore, we are reluctant to name a single parameter or cut-off for determining whether or not to attempt the evaluation of a given cCTA study.

As our protocol is in routine use at our institution and ICA was omitted in a significant proportion of patients, the shown data suggest that patients without an additional ICA (Group B) have a comparable number of procedure- and contrast-related events, consistent with the findings of Chieffo et al. [24]. AKI, as well as cardiovascular mortality, were slightly lower, and no myocardial infarction occurred in Group B. Despite the lack of sufficient power, this suggests that the chosen practice of using cCTA as a screening tool for CAD is safe and may help to further improve the safety of TAVI, to expedite treatment and reduce cost related to additional invasive diagnostics, complications and prolonged hospital stay.

Our study demonstrates that cCTA can be incorporated into the standard planning-CT for pre-TAVI evaluation without the need for additional contrast medium or medication. The chosen scan protocol is efficient and has a high rate of technical success despite this very challenging patient cohort, being only marginally influenced by patient characteristics (BMI) and scan demographics (heart rate).

The general capability of cCTA to exclude CAD in this high-risk group of patients pre-TAVI has been examined in several studies with promising results (Table 7) [13,14,15,16,17,18,19,20,23]. The advantages of cCTA with its high sensitivity and a particularly high NPV could be preserved in most of these studies and are being noted in current guidelines [1,5,6,7] (Figure 4 and Figure 5). However, the poor assessment of degree when a stenosis is detected with ultimately low positive predictive value remain an expected methodical limitation of cCTA.

Therefore, we tried to validate these results in our study by choosing a representative cohort, consisting of all patients that were examined during the investigated interval (except for prior CABG), purposefully including patients with arrhythmias, severe obesity and known CAD.

Since TAVI is becoming a viable alternative to surgery for even younger patients with hence fewer calcifications [3,4], the importance of cCTA during pre-TAVI evaluation is expected to rise even further.

Several limitations must be named. Firstly, the study design was retrospective and performed at a single center. Patients with prior CABG were not considered for this analysis. However, as they constituted only a small proportion of our cohort and are known to be very well evaluable with cCTA, we believe our results would not have been altered if this subgroup had been included [14,18]. In addition, patients with prior CABG may have complicated the evaluation of clinical events, while not being sufficient in number for meaningful subgroup analyses.

A diameter-based stenosis of ≥ 50% will not be a relevant cut-off for treatment in most cases (particularly not for older patients with severe aortic stenosis). Yet, this is the cut-off recommended by the SCCT for the analysis of cCTA to decide whether or not to recommend further workup or not and being used in most CT studies [28]. Therefore, we deliberately chose this cut-off to utilize cCTA’s strengths (high sensitivity and high NPV). Having opted for the clinically more relevant cut-off of 70% or even 90% used in ICA is very difficult in cCTA for technical reasons and therefore would have significantly decreased both sensitivity and NPV and thereby diminished the capacity of cCTA to serve as a reliable screening test. Instead of changing the threshold for CAD on cCTA, one could suggest a stepwise diagnostic approach to improve specificity with non-invasive functional testing like MRI or scintigraphy or via CT-derived fractional flow reserve, which does not require an additional examination [38,39,40].

Additionally, the decision to undertake further diagnostic workup with ICA partly depending on the diagnosis of cCTA causes both a verification and spectrum bias and enriched the verification cohort with more challenging cases and does therefore not lead to overestimation of the test’s performance in regard to NPV. Despite these limitations, the relatively large group of otherwise unselected patients should yield realistic expectations of this diagnostic approach.

Furthermore, cCTA and ICA were not always performed in immediate temporal proximity, with a gap between the two examinations of up to 3 months. However, as cCTA is a very sensitive test, the median time gap was 22.0 (8.7–108.5) hours and no relevant clinical events occurred between the two examinations, we do not believe that the quality of our results was affected by this circumstance.

Lastly, sample size, particularly of Group B, was relatively small and as the decision not to perform ICA when cCTA was negative was a clinical decision and not random, the expected health benefit of reduced complications in Group B could not be demonstrated during the period of investigation.

One should note that the protocol with dual acquisition used for our analysis requires a dual-source scanner and therefore may not be utilized in all institutions.

## 5. Conclusions

cCTA can be incorporated into pre-TAVI CT-evaluation with considerable technical success and without increasing the amount of contrast medium needed. cCTA may allow for exclusion of significant CAD in this high-risk patient group and may be valuable even in patients with moderate to high CAC or stents. Thereby, cCTA has the potential to decrease the need for an additional ICA in this co-morbid patient population.

## Figures and Tables

**Figure 1 jcm-09-01623-f001:**
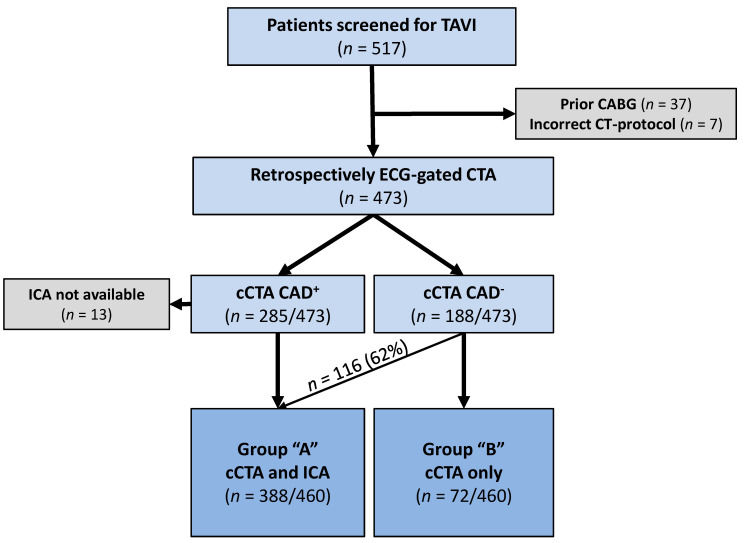
Flow-chart of the study population according to diagnostics received: Group A underwent cCTA and ICA; Group B underwent cCTA only. CABG = coronary artery bypass graft, CAD^−^ = no significant CAD on cCTA, CAD^+^ = significant CAD (stenosis ≥ 50%) on cCTA, cCTA = coronary CT-angiography, ICA = invasive coronary angiography, PCI = percutaneous coronary intervention, TAVI = transcatheter aortic valve implantation.

**Figure 2 jcm-09-01623-f002:**
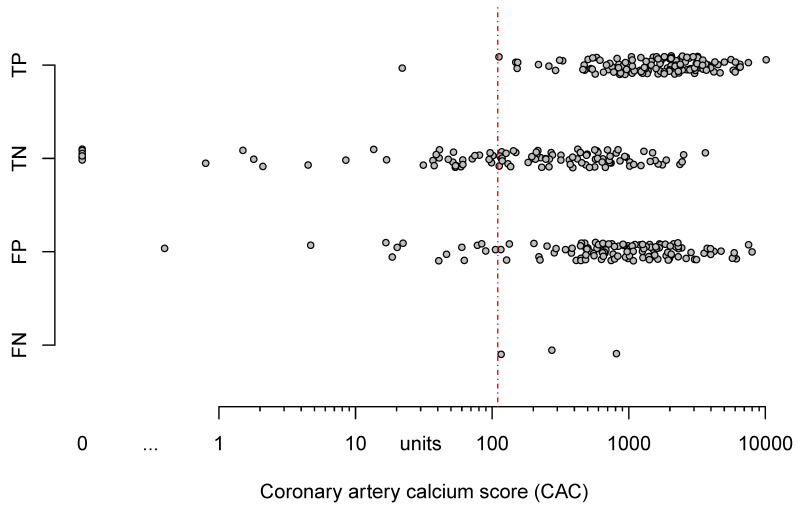
Dot-plot of patient’s CAC and their classification according to cCTA against invasive coronary angiography with QCA. Note the vast overlap of true negative and false positive results in relationship to CAC. The threshold (red dashed line) drawn at a CAC of 110 would exclude all false negative results as well as most true negative results. CAC = coronary artery calcium score, FN = false negative, FP = false positive, TN = true negative, TP = true positive, QCA = quantitative coronary analysis.

**Figure 3 jcm-09-01623-f003:**
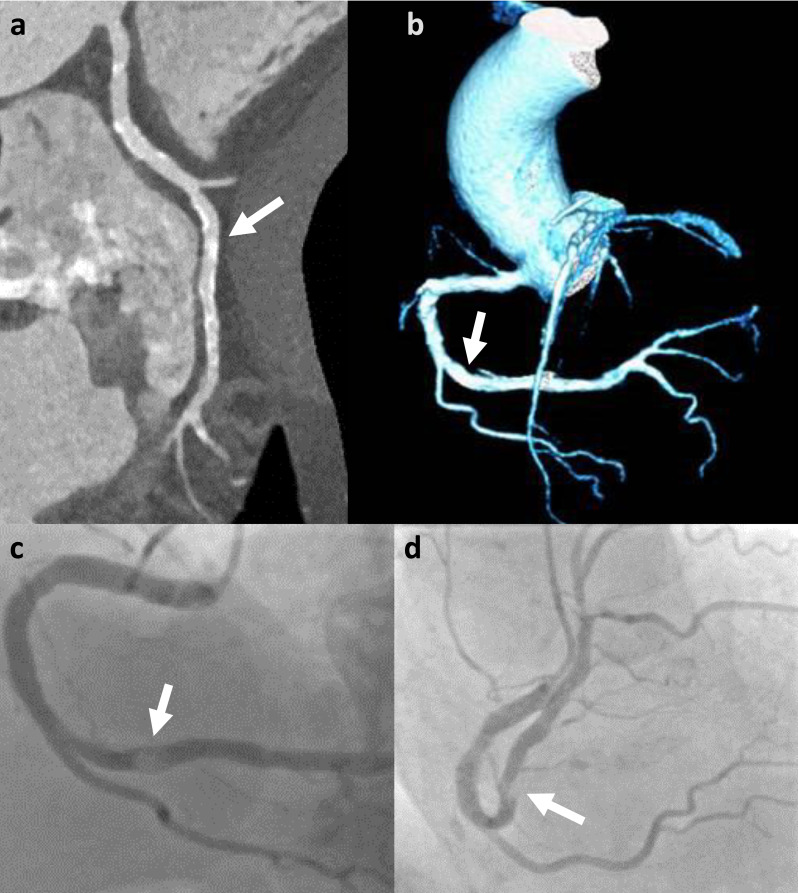
False negative (per vessel) cCTA example (CAC = 2839): Right coronary artery (RCA) with heavy calcifications and excellent contrast opacification (approx. 600 HU) on cCTA with multiplanar reconstruction (**a**) and volume rendered technique (**b**) and invasive coronary angiography depicting a ≥ 50% stenosis in the distal RCA (segment 3) (arrows) (**c**,**d**), which is masked on cCTA. CAC = coronary artery calcium score.

**Figure 4 jcm-09-01623-f004:**
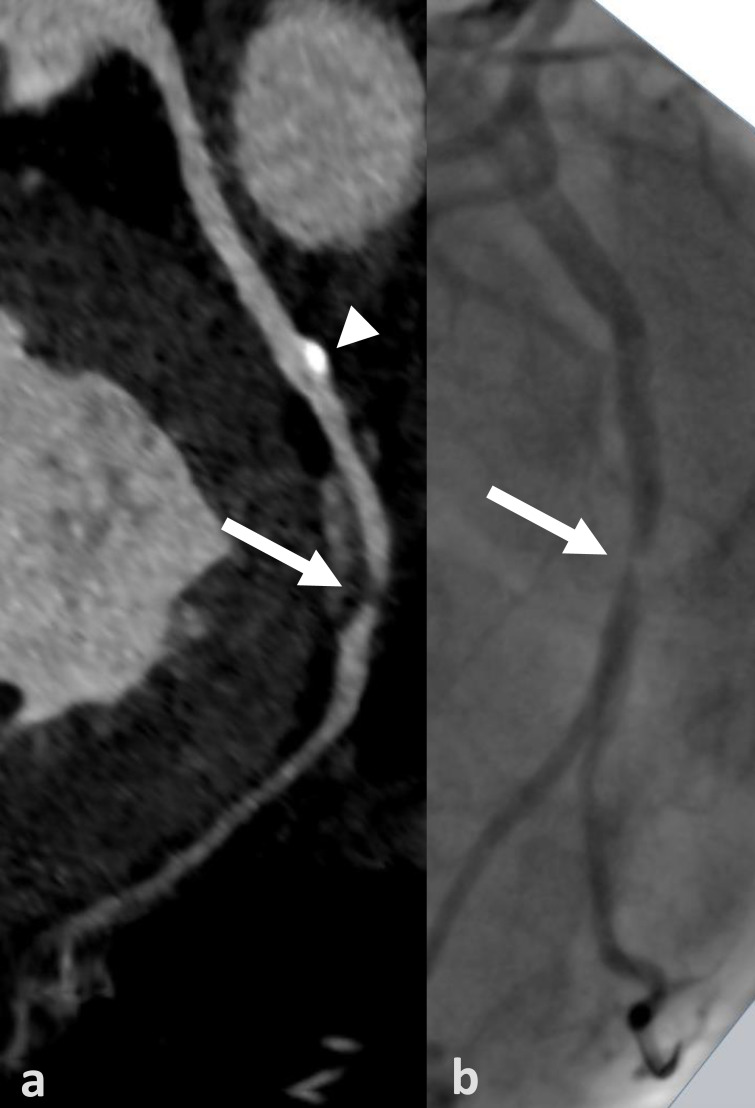
True positive cCTA example (CAC = 1235): Curved multiplanar reconstruction of the left circumflex artery (LCX) with non-calcified plaque and stenosis of 70% (arrow) in the distal LCX (segment 13) and a calcified plaque (arrowhead) in the proximal LCX (segment 11) (**a**). Corresponding projection of the invasive coronary angiography well depicting the stenosis (arrow); note the calcified non-stenotic plaque is not visible (**b**). CAC = coronary artery calcium score.

**Figure 5 jcm-09-01623-f005:**
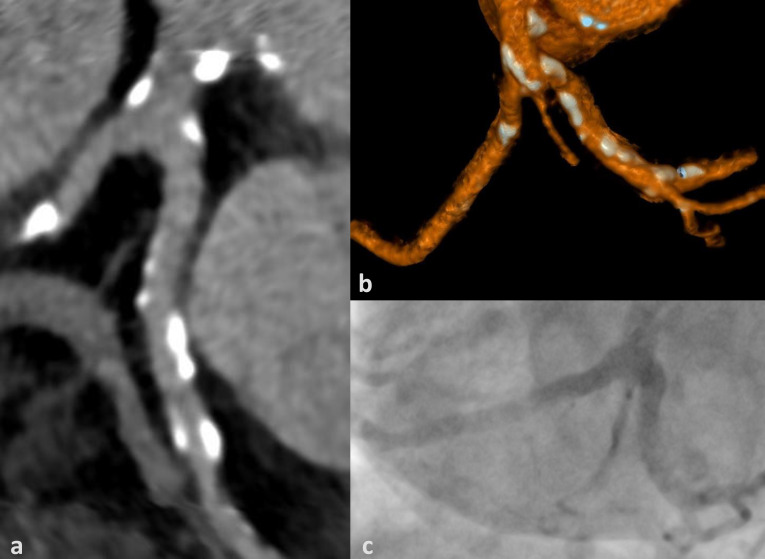
True negative cCTA example (CAC = 1834): Heavily calcified trifurcation of the left main (LM) into left anterior descending (LAD), left circumflex (LCX) and intermediate artery without luminal obstruction depicted as curved multiplanar reformation (**a**), volume rendered technique (**b**) and corresponding projection of invasive coronary angiography (**c**). CAC = coronary artery calcium score.

**Table 1 jcm-09-01623-t001:** Baseline characteristics.

	Group A		Group B		*p* Value
	*n* = 388		*n* = 72		
Age (years)	79.6	±7.2	79.9	±8.1	0.75
Female	191	49.2%	47	65.3%	0.01
BMI (kg/m^2^)	29.2	±6.2	29.3	±5.2	0.87
Diabetes mellitus (Type I or II)	49	12.6%	11	15.3%	0.55
Hypertension	346	89.2%	64	88.9%	0.74
Hyperlipidemia	227	58.5%	38	52.8%	0.13
Chronic kidney disease	164	42.3%	34	47.2%	0.58
Prior PCI	111	28.6%	4	5.6%	<0.001
Prior myocardial infarction	49	12.6%	1	1.4%	0.005
Peripheral artery disease	43	11.1%	7	9.7%	0.74
Prior stroke or TIA	31	8.0%	5	6.9%	0.77
Current smoking	51	13.1%	7	9.7%	0.36
NYHA classification III/IV	260	67.0%	37	51.4%	0.006
Left ventricular EF (%)	55.3	±12.9	59.9	±12.7	0.003
Aortic valve area (cm^2^)	0.70	(0.60–0.90)	0.70	(0.60–0.90)	0.39
Logistic EuroSCORE (%)	15.2	(10.1–23.7)	13.0	(7.8–19.6)	0.07
HR at rest (beats/min)	76.8	±15.4	75.6	±15.7	0.38
Sinus rhythm	215	64.7%	49	68.1%	0.48

Data are mean ± standard deviation, median and interquartile range in parenthesis or count and frequency in percent. BMI = body mass index; EF = ejection fraction, EuroSCORE = European System for Cardiac Operative Risk Evaluation [32], HR = heart rate, NYHA = New York Heart Association, PCI = percutaneous coronary intervention, TIA = transient ischemic attack.

**Table 2 jcm-09-01623-t002:** Scan demographics.

	Group A		Group B		*p* Value
	*n* = 388		*n* = 72		
HR during scan (beats/min)	71.5	(62.5–83.1)	71	(60.5–81.9)	0.5
HR variability (beats/min)	8	(2.0–37.0)	5.5	(2.0–34.0)	0.34
Contrast to noise ratio (HU)	11.6	±4.3	11.9	±4.1	0.59
CAC	859	(285–1975)	99	(36–303)	<0.001
	CAD^+^	CAD^−^			<0.001
	*n* = 272	*n* = 116			
	1792	463			
	(617–2330)	(60–628)			
Tube potential 70 kV	8	2.1%	2	2.8%	0.07
Tube potential 80 kV	213	54.9%	34	47.2%	
Tube potential 100 kV	139	35.8%	34	47.2%	
Tube potential 120 kV	18	4.6%	0	0%	
DLP (mGy×cm)	900.4	±384.9	834.1	±317.8	0.12

Data are mean ± standard deviation, median and interquartile range in parenthesis or count and frequency in percent. CAC = coronary artery calcium score, CAD− = negative for coronary artery disease, CAD+ = positive for coronary artery disease, DLP = Dose length product, HR = heart rate.

**Table 3 jcm-09-01623-t003:** Qualitative imaging parameters of cCTA per patient in Group A.

Image Quality Parameter on cCTA − Group A	cCTA CAD^−^		cCTA CAD^+^		*p* Value
	*n* = 116		*n* = 272		
Mean Calcification Score (0–3)	0.74	±0.67	1.81	±0.97	<0.001
Mean Artifact Score (0–3)	0.41	±0.62	0.90	±1.06	<0.001
Mean Opacification Score (0–3)	2.36	±0.75	2.18	±0.80	0.05
Mean Image Quality (0–3)	2.17	±0.74	1.69	±0.97	<0.001

Data are mean ± standard deviation. CAD^−^ = negative for significant coronary artery disease; CAD^+^ = positive for significant CAD.

**Table 4 jcm-09-01623-t004:** Diagnostic performance of cCTA in comparison to ICA with QCA.

	*n*	TP	TN	FP	FN	Sen.	Spe.	PPV	NPV
Coronary segments	4947	222	3989	702	34	86.7%	85.0%	24.0%	99.2%
Coronary vessels	1551	189	980	366	16	92.2%	72.8%	34.1%	98.4%
Patients with stents	87	45	9	33	0	100.0%	21.4%	57.7%	100.0%
All patients	388	135	113	137	3	97.8%	45.2%	49.6%	97.4%

FN = false negative, FP = false positive, NPV = negative predictive value, PPV = positive predictive value, Prev. = prevalence, Sen. = sensitivity, Spe. = specificity, TN = true negative, TP = true positive.

**Table 5 jcm-09-01623-t005:** Multiple logistic regression models of baseline characteristics and scan demographics and accuracy of cCTA.

	Odds Ratio	95% Confidence Interval	*p* Value
Model 1 a			
BMI	0.97	[0.93–1.00]	0.05
HR	0.99	[0.98–1.00]	0.11
Model 1 b			
BMI (5 kg/m^2^)	0.82	[0.69–0.99]	0.03
Model 2			
BMI	0.97	[0.93–1.01]	0.11
HR	0.99	[0.98–1.00]	0.09
_log_CAC	1.64	[0.35–7.65]	0.52
√ number of calcified lesions	0.84	[0.55–1.28]	0.40
_log_CAC/ number of calcified lesions	0.49	[0.12–2.05]	0.32

BMI = body mass index; CAC = BMI = body mass index; CAC = coronary artery calcium score, HR = heart rate.

**Table 6 jcm-09-01623-t006:** In-hospital events of patients having undergone TAVI.

In Hospital Events	Group A *n* = 273		Group B *n* = 58		*p* Value
MACCE	17	6.2%	4	6.9%	0.56
All-cause mortality	7	2.6%	1	1.7%	0.72
Cardiovascular mortality	3	1.1%	0	0.0%	0.48
Cerebrovascular events	12	4.4%	3	5.2%	0.81
Myocardial infarction	2	0.7%	0	0.0%	0.59
Acute kidney injury	23	8.4%	4	6.9%	0.71

Data are count and frequency in percent. MACCE = major adverse cardiovascular and cerebrovascular events, defined as the sum of all-cause mortality, cerebrovascular events, and myocardial infarction with a maximum of one event per patient [31].

**Table 7 jcm-09-01623-t007:** Diagnostic performance of cCTA per patient for detecting relevant CAD in the literature.

	*n*	Prev.	TP	FP	TN	FN	Sen.	Spe.	PPV	NPV
Pontone et al. (2011) [19]	60	43.3%	23	4	30	3	88.5%	88.2%	85.2%	90.9%
Andreini et al. (2014) [13]	325	29.8%	87	21	207	10	89.7%	90.8%	80.6%	95.4%
Hamdan et al. (2015) [15]	115	42.6%	47	18	48	2	95.9%	72.7%	72.3%	96.0%
Harris et al. (2015) [16]	100	74.0%	73	11	15	1	98.6%	57.7%	86.9%	93.8%
Opolski et al. (2015) [18]	475	56.8%	265	129	76	5	98.1%	37.1%	67.3%	93.8%
Matsumoto et al. (2017) [17]	60	40.0%	22	15	21	2	91.7%	58.3%	59.5%	91.3%
Rossi et al. (2017) [20]	140	41.4%	53	37	45	5	91.4%	54.9%	58.9%	90.0%
Annoni et al. (2018) [14]	115	20.0%	22	12	80	1	95.7%	87.0%	64.7%	98.8%
Strong et al. (2019) [23]	200	34.5%	69	76	55	0	100.0%	42.0%	47.6%	100.0%
Our results	388	35.6%	135	137	113	3	97.8%	45.2%	49.6%	97.4%
Combined results	1978	41.9%	40.2%	23.3%	34.9%	1.6%	96.1%	60.0%	63.4%	95.6%

FN = false negative, FP = false positive, NPV = negative predictive value, PPV = positive predictive value, Prev. = prevalence, Sen. = sensitivity, Spe. = specificity, TN = true negative, TP = true positive.

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
