# Peer review of "Combined Coronary CT-Angiography and TAVI-Planning: A Contrast-Neutral Routine Approach for Ruling-Out Significant Coronary Artery Disease"

_jcm, 2020, doi:10.3390/jcm9061623_

Round 1

Reviewer 1 Report

Thank you very much for the opportunities to review this manuscript.

Robin F et al. analyzed the ability of pre-procedural cCTA to rule out CAD in 460 patients with aortic stenosis who undergo TAVI. Data was retrospectively analyzed in detail. The manuscript is well written in logical and academic manner. As authors pointed out in the manuscript, there are literatures with similar topic previously. However, in this field, the role and future advantage of cCTA is still important and impactful for readers.

The reviewer has some minor comments on the manuscript.

  1. Abstract is well written, but please consider to add the definition of CAD in this study. It will help for readers to understand the story.
  2. Introduction: minor spell check required. L58: W → who. 
  3. methods: even in retrospective study, IRB approval number, confront of declaration of Helsinki or STROBE should be described in this section according to the ethical manner of clinical research. 
  4. Results: well written. The reviewer does not have any additional comments.
  5. Discussion: well written, but please discuss about 1) the significance and impact of CAD in patients with AS before TAVR, 2)   is a diameter stenosis of >50% proper cut-off value as significant CAD for AS patients? and 3) future and potential role of physiological assessment of CAD before TAVR with FFR, iFR or RI (Instantaneous Wave-Free Ratio for the Assessment of Intermediate Coronary Artery Stenosis in Patients With Severe Aortic Valve Stenosis. Comparison With Myocardial Perfusion Scintigraphy. J Am Coll Cardiol Intv 2018;11: 2032-2040.).

Author Response

Response to Reviewer 1 Comments

Point I: Robin F et al. analyzed the ability of pre-procedural cCTA to rule out CAD in 460 patients with aortic stenosis who undergo TAVI. Data was retrospectively analyzed in detail. The manuscript is well written in logical and academic manner. As authors pointed out in the manuscript, there are literatures with similar topic previously. However, in this field, the role and future advantage of cCTA is still important and impactful for readers.

Response I: We would like to thank the reviewer for supporting our article to be published in the Journal of Clinical Medicine and we feel truly honored to receive such positive response.

Point 1: Abstract is well written, but please consider to add the definition of CAD in this study. It will help for readers to understand the story.

Response 1: Thank you for your remark. We have added a short definition of CAD into the last sentence of the abstract’s background.

Change: L: 35/35(stenosis ≥50%)

Point 2: Introduction: minor spell check required. L58: W → who.

Response 2: Thank you for pointing out this error. We have corrected it. Additionally, we took the liberty to correct some minor typographical errors.

Point 3: methods: even in retrospective study, IRB approval number, confront of declaration of Helsinki or STROBE should be described in this section according to the ethical manner of clinical research. 

Response 3: Thank you for this important clarification. The IRB approval number is referred to in our manuscript as “(reference number: 435/18-ek)” and the respective section in the end of 2.1 Study design and patient population (L98-99) has been expanded by a statement regarding the Declaration of Helsinki.

Changes: L: 102-104/102-104 “This study was conducted in compliance with the Declaration of Helsinki (Medical Association 2013). The local ethics committee approved the study and written informed consent was waived (reference number: 435/18-ek).

Point 4: Results: well written. The reviewer does not have any additional comments.

Response 4: Thank you for the acknowledgment and this kind remark.

Point 5: Discussion: well written, but please discuss about 1) the significance and impact of CAD in patients with AS before TAVR,

2) is a diameter stenosis of >50% proper cut-off value as significant CAD for AS patients? and

3) future and potential role of physiological assessment of CAD before TAVR with FFR, iFR or RI (Instantaneous Wave-Free Ratio for the Assessment of Intermediate Coronary Artery Stenosis in Patients With Severe Aortic Valve Stenosis. Comparison With Myocardial Perfusion Scintigraphy. J Am Coll Cardiol Intv 2018;11: 2032-2040.).

Response 5: Thank you for these thoughtful remarks.

1) This aspect has been added to the Discussion

Change: L: 274-281/275-282 “The impact of CAD on outcome after TAVI is unclear and remains a controversial topic in itself – being still actively debated. Nevertheless, diagnosis of CAD prior to TAVI remains part of the routine pre-TAVI work-up for several reasons, including the overlap of risk factors and symptomatology between both disease entities, and the necessity to exclude and potentially treat severe proximal disease prior to TAVI according to current guideline recommendations [1,6]. Regardless of TAVI being an established procedure for the treatment of severe aortic stenosis, the experience of excluding CAD with cCTA in this patient group is still limited with less than 2.000 patients reported on so far.

2) This aspect has been added to the Discussion

Change: L: 362-369/364-371 “A diameter-based stenosis of 50% will in most cases not be a relevant cut-off in regard to treatment (particularly not for older patients with severe aortic stenosis). Yet, this is the cut-off recommended by the SCCT for the analysis of cCTA to decide whether or not to recommend further workup or not and being used in most CT studies [28]. Therefore, we deliberately chose this cut-off to utilize cCTA’s strengths (high sensitivity and high NPV). Having opted for the clinically more relevant cut-off of 70% or even 90% used in ICA is very difficult in cCTA for technical reasons and therefore would have significantly decreased both sensitivity and NPV and thereby diminished the capacity of cCTA to serve as a reliable screening test.

3) Using invasive functional testing instead of “only” QCA as the standard of reference may have improved the clinical message further. Yet severe aortic stenosis may change the hemodynamics of patients, in turn influencing the results of invasive functional testing and therefore potentially requiring new cut-offs. As this is an even more complex topic and we do not have invasive measurements available for most our patients, we do not believe adding this aspect would improve the message of our manuscript.

Coronary Hemodynamics in Patients With Severe Aortic Stenosis and Coronary Artery Disease Undergoing Transcatheter Aortic Valve Replacement: Implications for Clinical Indices of Coronary Stenosis Severity. JACC Cardiovasc Interv. 2018 Oct 22;11(20):2019-2031. doi: 10.1016/j.jcin.2018.07.019. Epub 2018 Aug 25.

Physiologic evaluation of coronary lesions using instantaneous wave-free ratio (iFR) in patients with severe aortic stenosis undergoing transcatheter aortic valve implantation. EuroIntervention. 2018 Jan 20;13(13):1512-1519. doi: 10.4244/EIJ-D-17-00542.

For these Reasons we decided to give an outlook in the last part of Discussion:

Change: L: 369-372/371-374: “Instead of changing the threshold for CAD on cCTA, one could suggest a stepwise diagnostic approach to improve specificity with non-invasive functional testing like MRI or scintigraphy or via CT-derived fractional flow reserve, which does not require an additional examination [38–40].

Reviewer 2 Report

The stated objective of the study was to assess the accuracy and safety of coronary CT angiography (CCTA) for the screening of significant coronary artery disease (CAD) in patients who were being assessed for transcatheter aortic valve implantation (TAVI).

The strengths of the study include: the important clinical question [whether CCTA can be accurate enough to serve as a screening tool for significant CAD and thus obviate the need for invasive coronary angiography (ICA) in TAVI candidates]; the inclusion of patients with both CCTA and the gold-standard ICA for comparison; and certain elements of the methodology (such as CCTA interpretation blinded to the ICA results; analysis of test characteristics of CCTA only within the group of patients with both CCTA and ICA).

However, there are several major and minor issues outlined below that must be addressed before further consideration for publication. The main limitation of the study is the lack of novelty. Several prior studies have shown a high sensitivity and negative predictive value of CCTA in identifying significant coronary artery disease, including within the TAVI population, as the authors themselves summarize in Table 7. Whether or not to accept this manuscript for publication will depend on whether the Editors believe there is sufficient incremental value and interest in the study results for the readers of this particular Journal.

Major issues:

  1. As mentioned, this study lacks novelty, which is a major limitation of the study. Several of the main study findings have been shown by prior studies including: the high sensitivity and negative predictive value of CCTA in identifying significant CAD; the negative effects of calcification, high BMI, and heart rate in reducing the accuracy of CCTA; and the potential of CCTA to reduce the amount of contrast exposure and risk of acute kidney injury (AKI) by obviating the need for ICA in select patients. 
  2. The one possible claim to novelty is the inclusion of patients with stents, high heart rate, arrhythmia, elevated body mass index (BMI), or suboptimal contrast opacification. Table 4 provides an interesting subgroup analysis of the diagnostic performance of CCTA among patients with stents. Could the authors perform and present the results of similar subgroup analyses in those other subgroups of patients with technically challenging studies?
  3. The authors are commended for attempting to demonstrate a difference in outcomes (in the form of in-hospital events) in patients who underwent ICA in addition to CCTA compared to those who underwent CCTA only (Table 6). However, the results of Table 6 are misleading for a number of reasons and should either be modified or omitted entirely. First, not all of the patients who underwent pre-TAVI assessment eventually proceeded with the TAVI. Therefore, the outcomes presented are of a subgroup of patients with a different set of baseline characteristics than was presented in Table 1. Excluding patients who did not eventually undergo TAVI would have resulted in a a more appropriate cohesive analysis throughout the manuscript with patient characteristics listed in Table 1 matching the same patients whose outcomes are listed in Table 4. Second, are "in-hospital events after TAVI" the appropriate outcomes to examine for the purposes of this study? If the objective of the study was to assess safety of CCTA as a screening tool in place of ICA, then should the outcome not be rates of AKI both during pre-TAVI assessment and post-TAVI? Third, the comparison between Group A and Group B does not seem appropriate given the significant intrinsic differences between the two groups. Group A consists of a large proportion of patients with significant CAD who are at higher risk for CAD-related events than those in group B. In addition, wow many of these patients underwent PCI and therefore received additional contrast? A slightly better comparison could be between those patients in Group A with a negative CCTA/ICA and patients in Group B in order to eliminate concomitant CAD as a confounding factor. However, since Group B consists of a highly selected group of patients who did not have significant CAD on CCTA but with impaired renal function who might be at greater risk for AKI, even this comparison poses a challenge. All of these baseline characteristics represent confounding variables that make it impossible to meaningfully interpret a comparison of outcomes between the two groups.
  4. The inclusion of Group B serves no purpose for the assessment of CCTA test characteristics since these patients did not undergo the gold standard ICA for comparison. The only apparent purpose of Group B was to compare safety and outcome measures between those who underwent additional ICA versus those who did not. However, this in itself is problematic, as described in #3 above. The authors should consider removing Group B entirely from the study.

Minor issues:

  1. The authors state in the Limitations section that they believe inclusion of patients with CABG would not have altered their results. Could the authors explain why these patients were therefore excluded?
  2. Some minor copy-editing issues identified throughout the manuscript:
  • Page 2, lines 57-58: change "wo" to "who" (twice in first sentence)
  • Page 2, line 80: change "perpetration" to "preparation"
  • Page 2, line 83: What is meant by "without any patient-specific adjustments"? Without excluding patients with technically challenging studies? Please either be more specific or consider omitting this phrase.
  • Page 3, line 99: change "waved" to "waived"

Author Response

Response to Reviewer 2 Comments

Point I: The stated objective of the study was to assess the accuracy and safety of coronary CT angiography (CCTA) for the screening of significant coronary artery disease (CAD) in patients who were being assessed for transcatheter aortic valve implantation (TAVI).

The strengths of the study include: the important clinical question [whether CCTA can be accurate enough to serve as a screening tool for significant CAD and thus obviate the need for invasive coronary angiography (ICA) in TAVI candidates]; the inclusion of patients with both CCTA and the gold-standard ICA for comparison; and certain elements of the methodology (such as CCTA interpretation blinded to the ICA results; analysis of test characteristics of CCTA only within the group of patients with both CCTA and ICA).

However, there are several major and minor issues outlined below that must be addressed before further consideration for publication. The main limitation of the study is the lack of novelty. Several prior studies have shown a high sensitivity and negative predictive value of CCTA in identifying significant coronary artery disease, including within the TAVI population, as the authors themselves summarize in Table 7. Whether or not to accept this manuscript for publication will depend on whether the Editors believe there is sufficient incremental value and interest in the study results for the readers of this particular Journal.

Response I: We would like to thank the reviewer for acknowledging and pointing out the strengths of our work and giving us such detailed feedback on how we may improve our work even further.

Point 1: As mentioned, this study lacks novelty, which is a major limitation of the study. Several of the main study findings have been shown by prior studies including: the high sensitivity and negative predictive value of CCTA in identifying significant CAD; the negative effects of calcification, high BMI, and heart rate in reducing the accuracy of CCTA; and the potential of CCTA to reduce the amount of contrast exposure and risk of acute kidney injury (AKI) by obviating the need for ICA in select patients.

Response 1: Indeed, many of the aspects reported on in our study have previously been reported in several separate studies and in many cases in different patient cohorts. Yet, our study is novel in its comprehensiveness, revealing that those findings may also be observed in patients prior to TAVI. We then go on to analyze the influence of these mentioned factors on the validity of cCTA, and we found that many of these factors/ challenges (high heart rate, arrhythmias, CAC) in these patients are much less of an obstacle than previously suggested or assumed with the technical improvements currently available.

Furthermore and as acknowledged by the reviewer, please note that many of the studies cited regarding coronary CT-evaluation in TAVI-patients have made important exclusions regarding challenging patients and their results are therefore much more difficult to generalize to one’s individual practice.

Lastly, we integrated coronary analysis into our protocol without having had to increase the amount of contrast medium applied for CT.

We therefore feel that our study is sufficiently novel in its approach and attempt to give a more general approach to this topic.

Changes:

We added the following sentences to the Introduction

L: 78-82/78-82 “Regardless of this and despite TAVI being an established procedure with 24.808 annual entries into the STS/ACC TVT Registry in 2015, an estimated 107.000 TAVI prosthesis sold in 2017 and 971 procedures performed in 2019 in our institution alone, the experience in evaluation of CAD with cCTA is still very limited with less than 2.000 patients reported on so far [25,26].

and discussion

L:279-281/280-282: “Regardless of this and despite TAVI being an established procedure with 24.808 annual entries into the STS/ACC TVT Registry in 2015, an estimated 107.000 TAVI prosthesis sold in 2017 and 971 procedures performed in 2019 in our institution alone, the experience in evaluation of CAD with cCTA is still very limited in this cohort with less than 2.000 patients reported on so far [25,26].”.

Point 2: The one possible claim to novelty is the inclusion of patients with stents, high heart rate, arrhythmia, elevated body mass index (BMI), or suboptimal contrast opacification. Table 4 provides an interesting subgroup analysis of the diagnostic performance of CCTA among patients with stents. Could the authors perform and present the results of similar subgroup analyses in those other subgroups of patients with technically challenging studies?

Response 2: Thank you for this thoughtful suggestion. We employed multiple logistic regression analyses to explore the important relationship of potentially interfering factors and validity of cCTA analysis and present the results in Table 5. A simpler analysis regarding the more subjective grading of image quality is presented in Table 3. We fear that presenting the results in the suggested fashion would not add much additional information and could without a significance test potentially be potentially misleading, as our sample size remains moderate.

Point 3: a) The authors are commended for attempting to demonstrate a difference in outcomes (in the form of in-hospital events) in patients who underwent ICA in addition to CCTA compared to those who underwent CCTA only (Table 6). However, the results of Table 6 are misleading for a number of reasons and should either be modified or omitted entirely. First, not all of the patients who underwent pre-TAVI assessment eventually proceeded with the TAVI. Therefore, the outcomes presented are of a subgroup of patients with a different set of baseline characteristics than was presented in Table 1. Excluding patients who did not eventually undergo TAVI would have resulted in a a more appropriate cohesive analysis throughout the manuscript with patient characteristics listed in Table 1 matching the same patients whose outcomes are listed in Table 4.

  1. b) Second, are "in-hospital events after TAVI" the appropriate outcomes to examine for the purposes of this study? If the objective of the study was to assess safety of CCTA as a screening tool in place of ICA, then should the outcome not be rates of AKI both during pre-TAVI assessment and post-TAVI?
  2. c) Third, the comparison between Group A and Group B does not seem appropriate given the significant intrinsic differences between the two groups. Group A consists of a large proportion of patients with significant CAD who are at higher risk for CAD-related events than those in group B. In addition, wow many of these patients underwent PCI and therefore received additional contrast? A slightly better comparison could be between those patients in Group A with a negative CCTA/ICA and patients in Group B in order to eliminate concomitant CAD as a confounding factor. However, since Group B consists of a highly selected group of patients who did not have significant CAD on CCTA but with impaired renal function who might be at greater risk for AKI, even this comparison poses a challenge. All of these baseline characteristics represent confounding variables that make it impossible to meaningfully interpret a comparison of outcomes between the two groups.

Response 3: a) Thank you for acknowledging the effort it took to collect the data needed for this sub-analysis. Point number a) is definitely a valid one. However, we believe that defining the sensitivity and specificity of cCTA compared to ICA is different from looking at outcomes of patients eventually undergoing the TAVI procedure and have therefore opted for this differential analysis.

  1. b) This is also a valid point and we actually performed our analysis the way the reviewer suggests. We therefore revised the description of Table 6 (now reads: “In-hospital events of patients having undergone TAVI”) and clarified the methods section to describe the approach used in our manuscript more clearly. The section now reads L: 159-160/159-160 “MACCE in addition to acute kidney injury (AKI) were documented for the duration of the entire hospital stay…”.
  2. c) This is a great observation made and we do agree with the reviewer’s opinion. However, it is unlikely to show differences by excluding a significant proportion (>200 patients) of Group A from this analysis. In addition, we are uncertain if reducing the size of our sample so drastically would outweigh the benefit of excluding the mentioned important confounder of CAD. For these reasons, we decided to add this sub-group analysis into section 7. Group differences in MACCE and AKI L: 268-271/269-272 “Also for CAD- patients clinical events were not significantly different between Group A (n=73) and B (n=58). Overall MACCE were numerically more frequent in Group B (A: 1.4%; B: 6.9%;p=0.14), caused by cerebrovascular events (A: 1.4%; B: 5.2%;p=0.27) and all-cause mortality (A: 0.0%; B: 1.7%; p=0.43); AKI was numerically less frequent in Group B (A: 8.2%; B: 6.9%; p=0.77).

Point 4: The inclusion of Group B serves no purpose for the assessment of CCTA test characteristics since these patients did not undergo the gold standard ICA for comparison. The only apparent purpose of Group B was to compare safety and outcome measures between those who underwent additional ICA versus those who did not. However, this in itself is problematic, as described in #3 above. The authors should consider removing Group B entirely from the study.

Response 4: Thank you for raising this very important issue. Indeed, Group B did not undergo the standard of reference (ICA with QCA) after an already negative cCTA and therefore cannot serve to further characterize cCTA. As also acknowledged by the reviewer in #1, cCTA’s very high sensitivity and NPV are far from being a novelty and only not well characterized in this elderly population of TAVI-patients. Yet, in our approach omitting these patients from the entire analysis would significantly and misleadingly increase the prevalence of CAD in our supposed screening cohort. Furthermore, the important comparison of scan demographics (Table 2), that strikingly demonstrates no significant differences in many factors between these groups believed to be very important for technical success would not be possible.

Again and also in regard to the issues raised in #3, the most desirable approach of simply correlating cCTA in all manners against an invasive test, in our opinion, is ethically simply no longer acceptable, because of the overwhelming evidence regarding cCTA’s test characteristics and the risk it takes to verify a negative cCTA with an invasive test. Therefore, the only way to broaden and improve the knowledge in the field of cCTA and TAVI in its entirety will be through retrospective observations and registries with all their well known limitations, because it can no longer be justified to correlate a negative cCTA test with an invasive test for scientific purposes only.

Point minor 1: The authors state in the Limitations section that they believe inclusion of patients with CABG would not have altered their results. Could the authors explain why these patients were therefore excluded?

Response minor 1: We acquired the ECG-gated scan of the heart only and to serve a dual purpose –characterization of anatomy and dimensions relevant for TAVI and evaluation of coronary arteries – thereby, frequently not covering the entirety of eventual Coronary bypass grafts and particularly not LIMA- or RIMA-grafts. Furthermore, we decided on using QCA in two orthogonal views for the evaluation of ICA as the standard of reference. As grafts, particularly of the internal mammary arteries are frequently not depicted in two views in our practice. Although the origin of these grafts is less frequently diseased compared to the distal anastomosis, our standard of reference would have had to be adopted for these circumstances. Lastly, the considerations for such patients for TAVI in themselves are different because of altered coronary anatomy and physiology after having had all vessels grafted. This fact might influence the rate of clinical events in these patients.

Since the number of these patients is relatively low, prohibiting a meaningful subgroup analysis, and as stated in the Limitations, several previous studies [e.g. for TAVI-patients: 14,18] have already found the evaluation of coronary arteries and bypass grafts of patients after bypass surgery to be very reliable, we feel that the inclusion of this substantially different subgroup would have introduced more challenges in interpretation, rather than improving the clarity of our work and therefore made the conscious decision to exclude such patients for our analysis.

To reflect our opinion adequately in the manuscript, we have expanded the limitation by the following sentence.

Change: L: 359-361/360-363: “In addition, patients with prior CABG may have complicated the evaluation of clinical events because of changes in coronary artery physiology and anatomy and therefore differing considerations for TAVI, while not being sufficient in number for meaningful subgroup analyses.

Point minor 2: Some minor copy-editing issues identified throughout the manuscript:

Page 2, lines 57-58: change "wo" to "who" (twice in first sentence)

Page 2, line 80: change "perpetration" to "preparation"

Page 2, line 83: What is meant by "without any patient-specific adjustments"? Without excluding patients with technically challenging studies? Please either be more specific or consider omitting this phrase.

Page 3, line 99: change "waved" to "waived"

Response minor 2: Thank you for pointing out these errors. We have corrected them and specified the sentence.

L: 88/88 “In the present study we analyzed the accuracy and safety of cCTA acquired during pre-procedural CT as a primary screening tool for significant CAD in a large cohort of unselected TAVI patients without any patient-specific adjustments of scan parameters or patient preparation.”.

Furthermore, we took the liberty to correct some minor typographical errors.

We would like to thank both the editor and the reviewers for their kind consideration of our manuscript and the opportunity to revise our paper according to their suggestions.

We are looking forward to their esteemed analysis of our revision and hope that it may be found suitable for publication in the Journal of Clinical Medicine.

Most sincerely,

The authors

Round 2

Reviewer 2 Report

Thank you for addressing my comments. I have no further suggestions at this time.